# Influence of TiC Particles on Mechanical and Tribological Characteristics of Advanced Aluminium Matrix Composites Fabricated through Ultrasonic-Assisted Stir Casting

**Chitti Babu Golla [1], Mahammod Babar Pasha [1], Rajamalla Narasimha Rao [1,\*], Syed Ismail [1] and Manoj Gupta [2,\*]**

[1] Department of Mechanical Engineering, National Institute of Technology Warangal, Warangal 506004, India
[2] Department of Mechanical Engineering, National University of Singapore, 9 Engineering Drive 1, Singapore 117576, Singapore
\* Correspondence: rnrao@nitw.ac.in (R.N.R.); mpegm@nus.edu.sg (M.G.)

**Abstract:** The present investigation highlights the development of high-performance materials in the construction materials industry, with a special focus on the production of aluminium matrix composites (AMCs) containing titanium carbide (TiC) particles. The stir casting method with ultrasonic assistance was employed to enhance the mechanical and tribological properties. ASTM standards were employed to evaluate the influence of TiC particles on density, hardness (VHN), ultimate tensile strength (UTS), and wear resistance at various TiC weight fraction percentages (0.0 wt.%, 2.0 wt.%, 4.0 wt.%, 6.0 wt.%, and 8.0 wt.%). Field emission scanning electron microscopy (FESEM) and X-ray diffraction (XRD) analysis were performed to analyse the microstructural changes and elemental phases present in the synthesised composite. Results revealed that the incorporation of 8 wt.% TiC reinforcement in the metal matrix composites demonstrated significant improvements compared to the base alloy. In particular, a substantial enhancement in hardness by 32%, a notable increase of 68% in UTS, and a significant 80% rise in yield strength were observed when contrasted with the pure aluminium alloy. The tensile fracture analysis of the specimens revealed the presence of dimples, voids, and cracks, suggesting a brittle nature. To assess the wear characteristics of the composites, dry sliding wear experiments were performed using a pin-on-disc wear tester. Incorporation of TiC particles resulted in a lower coefficient of friction than the base alloy, with the lowest friction coefficient being recorded at 0.266 for 6 wt.% TiC, according to the data. FESEM and energy-dispersive X-ray spectroscopy (EDXS) were used to examine the surfaces of the worn pin. Overall, the inclusion of TiC reinforcement particles in the matrix alloy greatly enhanced the wear resistance and friction coefficient of the Al-6TiC composites. Ploughing and adhesion under lower loads and delamination under higher loads were the wear mechanisms observed in the wear test.

**Keywords:** aluminium; TiC; ultrasonic-assisted stir casting; wear resistance; wear mechanism

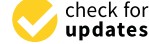



## 1. Introduction

In recent decades, metal matrix composites (MMCs) have emerged as new engineering materials with superior properties compared to monolithic materials. The favourable properties that are typically exhibited by metal matrix composites include light weight, high elastic modulus, high strength, high thermal stability, resistance to elevated temperatures, improved wear resistance, a low thermal expansion coefficient, and extended fatigue life [1,2]. Due to their exceptional blend of properties, advanced matrix composites (AMCs) have risen to prominence as advanced materials, playing a crucial role in diverse industries such as aerospace, automotive [3] (i.e., pistons and connecting rod [4]), defence, and various engineering sectors [5]. Aluminium [6–8], and magnesium are the usual matrices used for lightweight MMCs [9]. The most commonly used aluminium alloy (AA) matrix ranges from the AA-1xxx series to the AA-8xxx series. Among these, the AA-6xxx series and AA-7xxx series alloys are broadly researched because of their excellent corrosion

resistance, wear resistance, and high strength. Recently, AA8011 has become the preferred material for structural applications in modern engineering, including construction and transportation. This is due to its composition of Fe and Si alloying elements, which provide the necessary characteristics of good hardness, high tensile strength, and suitable wear resistance, which are crucial for such applications [10]. Despite its favourable properties, its use in automobile powertrain applications is limited by its poor wear resistance, which necessitates further improvement through the development of advanced composites [11]. To improve the low wear resistance of AA8011, researchers have studied the integration of hard secondary reinforcement phases into the aluminium matrix. Reinforcements such as carbides, borides, and oxides have been utilised for both ex situ and in situ composites [12]. The mechanical and surface properties of 8000 series aluminium alloys were enhanced through the incorporation of ceramic particles, such as $Al_2O_3$ [13], SiC [14,15], $TiO_2$ [16], $Si_3N_4/ZrB_2$ [17], $B_4C$ [18] and $TiB_2$ [19]. AMCs are presently manufactured using different methods such as mechanical alloying, additive manufacturing [20,21], stir casting [22], compo casting [23], powder metallurgy [24], squeeze casting [25], and spray deposition [26].

Liquid state processes have proven to be the most economical and productive approach for fabricating metal matrix composites. Their straightforward utilisation, economical processing cost, ability to scale up to large-scale production, and high output render them a preferred option. Stir casting, a typical liquid-state processing method for generating AMCs, has been combined with ultra-sonication in recent studies to produce efficient cavitation and ensure an even dispersion of reinforcement particles throughout the matrix [27]. Sakthivelu et al. [28] developed $AA8011/Al_2O_3$ composites by changing the weight percentage (4, 8, 12 wt.%) of the $Al_2O_3$ and investigated the composites' sliding wear behaviour. According to the results, composites were significantly more wear-resistant than base alloys. Siddharthan et al. [29] reported that SiC particle-reinforced AA8011 composites displayed improved resistance to wear. Likewise, Fayomi et al. [30] produced an AA8011 alloy composite with changing weight fractions of nano-$Si_3N_4$ using a two-step stir casting method. The authors concluded that the gradual presence of $Si_3N_4$ particles reduced the friction coefficient while increasing the resistance to wear. Moreover, associated with the base material, the hybrid AA8011 composites with $Si_3N_4$ and $ZrB_2$ reinforcements displayed much higher wear resistance. Keerthivasan et al. [31] demonstrated that the inclusion of TiC particles significantly enhances the mechanical behaviour of the composites, resulting in improvements in hardness and ultimate tensile strength. Baskaran et al. [32] observed that the integration of TiC particles into the AA7075 matrix leads to a substantial enhancement in wear resistance. Similarly, Siddappa et al. [33] developed AMCs based on Al6061 and TiC particulates (3%, 6%, 9%, and 12%) using the stir casting process. The results revealed a significant improvement in the hardness and tensile strength of the composites, with an enhancement of approximately 20% observed as the TiC particulate content progressively increased.

A comprehensive investigation has been carried out on various aluminium alloy metal matrix composites, including AA6061 [34], AA7075 [35], and AA2024 [36,37], as evident from the literature. However, there exists a limited amount of research focusing on the microstructural analysis, hardness, and wear characteristics of TiC-reinforced AA8011 matrix composites. As a result, the purpose of this research is to investigate how the presence of TiC particles affects the microstructure, hardness, and wear properties of AA8011 matrix composites. By addressing this research gap, valuable insights can be gained regarding the potential applications and performance of TiC-reinforced AA8011 composites. The outcomes of this investigation are likely to offer a deeper understanding of the material behaviour and pave the way for optimising the properties of AMCs in various engineering applications.

## 2. Materials and Methods

### 2.1. Materials

The present investigation utilised aluminium alloy (AA8011) as the base material (Table 1). It was procured from Bharat Aerospace Metals Pvt Ltd., Mumbai, India. Properties of AA8011 include density (2.7 g/cm$^3$), ultimate tensile strength (124 MPa), melting point (660 °C), modulus of elasticity (70.3 GPa), and thermal conductivity (237 W/m·k) [10].

**Table 1.** Chemical elements present in AA8011 aluminium alloy matrix (in wt.%).

| Element | Si | Fe | Ag | Mn | Cu | Mg | Ti | Zn | Al |
|---------|-----|-----|------|------|------|------|-------|-------|-----|
| AA8011 | 0.624 | 0.721 | 0.45 | 0.04 | 0.23 | 0.30 | 0.003 | 0.002 | Bal |

The TiC particles having an average particle size of 10 μm were procured from Parshwamani Metals, Mumbai, India. TiC particles have gained significant attention as reinforcement in AMCs due to their outstanding features, including a high melting temperature (3250 °C), high modulus of elasticity (450 GPa), high hardness (2470 Kg/mm$^2$), a density of 4.93 g/cm$^3$, a low coefficient of thermal expansion (7.61 μm/k), and excellent thermal conductivity (28.9 W/m·k). The TiC particle composition was revealed through a scanning electron micrograph (Figure 1a), showcasing particles with irregular shapes and various sizes. This variability in particle attributes arose due to the ex situ manufacturing process, which lacked precise control over size and shape. In (Figure 1b), the X-ray diffraction pattern of TiC is presented, exhibiting distinct peaks at (1 1 1), (2 0 0), (2 2 0), (1 1 3), and (2 2 2). These peaks confirm the nature of the recovered powder as TiC particles.

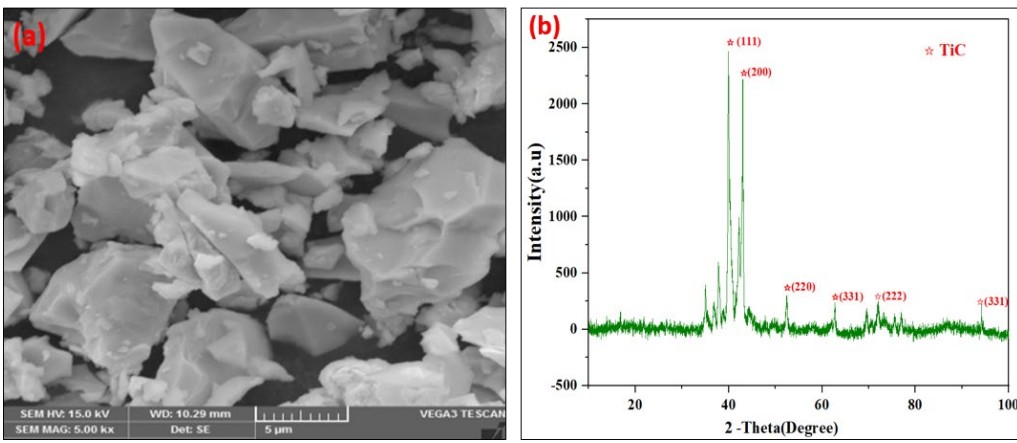

**Figure 1.** (**a**) Scanning electron micrograph of TiC particles. (**b**) X−ray diffraction analysis of TiC particles.

### 2.2. Fabrication of the AA8011-TiC Composites

In this study, an ultrasonically aided stir casting process was used to modify a traditional stir casting process for developing metal matrix composites. The AA8011 aluminium alloy matrix was melted in an electrical furnace at around 800 °C, which is higher than the melting temperature of pure aluminium. During the melting process, a mechanical stirrer was used to stir the molten metal. To improve the wettability of the AA8011 matrix, magnesium ribbons were used between the matrix and the TiC reinforcement. The molten surface was completely cleansed, and the molten metal's temperature was then raised to 750 °C. Preheated TiC microparticles were introduced to the liquid metal at various weight ratios, resulting in the formation of composite materials with different compositions. The stirring process was carefully maintained to ensure a homogeneous dispersion, with an average stirring speed of 350 rpm for approximately 15 min [38]. Afterward, the composite mixture was allowed to cool and reach a semi-solid state for 15 min. It was then reheated to 750 °C and subjected to an additional 5 min of stirring. The stirring speed, which was

controlled between 300 and 350 rpm, was constantly monitored. Ultrasonic-assisted stirring was used at this step, with a high-power ultrasonic generator with an input frequency of 20 kHz and an input power of 2 kW. An ultrasonic probe was placed in the composite blend for around 15 min, producing high-intensity ultrasonic sound waves [39]. Many dendrites with many branches might grow in the casting during the solidification process, reducing the mechanical properties of the composite. In order to remove this type of dendritic structure, an ultrasonic vibration was used during casting. The vibrations assist in breaking the dendritic structure and evenly distribute the particles in the casting [40]. This also enhances wettability and uniform dispersion of the TiC particles within the matrix material. Figure 2a–c, depicts Composite fabrication setup and the use of ultrasonic-assisted stir casting (Swap Equip, Chennai, India) for preparing composites with varying weight percentages of TiC (0%, 2%, 4%, 6%, and 8%). The same process was repeated to create composites with varying wt.% of TiC particles. The as-cast composites were machined into wear-testing shapes.

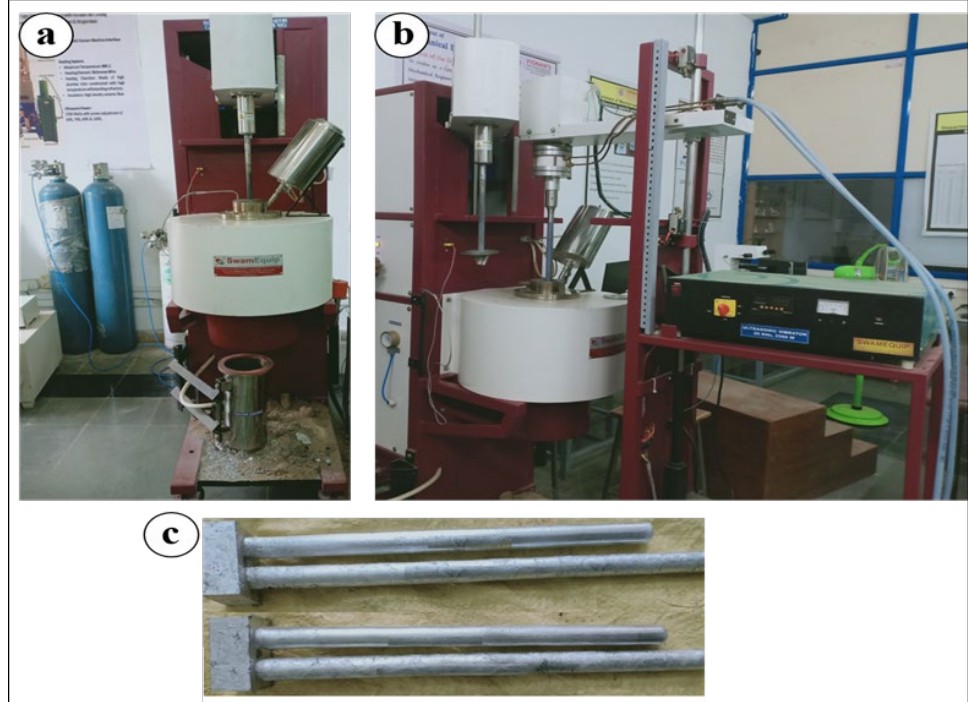

**Figure 2.** Composite fabrication setup: (**a**) Stir casting setup, (**b**) Ultrasonic setup attached to the stir casting, (**c**) Fabricated composite samples.

To obtain a fine surface finish, the machined samples were mechanically ground using silicon carbide abrasive grit papers ranging in grit size from 800 to 2000 and polished with diamond paste. To investigate the microstructure of the polished samples, etching was conducted using Keller's reagent, and observation was carried out using SEM and EDS. The samples' hardness was measured using a Vickers microhardness test as per ASTM E384-17 standard. The test involved creating 20 indentations spaced 0.5 mm apart and applying a 100 gf load for 15 s [12]. To determine the microhardness value of the samples, an average was calculated based on the 20 measurements obtained.

As illustrated in Figure 3, the dry sliding wear testing was carried out utilising a pin-on-disc testing device made by MAGNUM Instruments Pvt. Ltd., Bangalore, India. The testing procedure followed the established protocols outlined in the ASTM G99-09 standard. The cylindrical pin-shaped samples measured 10 mm in diameter and 30 mm in length. They are intended to slide against a counter disc made of EN31 steel, a common material in the manufacture of mechanical components. The Rockwell hardness scale C specifies EN31 steel as having a hardness of 60 ± 3 HRC. As a result, the pin samples

were designed to be subjected to a sliding test against a material of a specific hardness. Experiments were conducted in a room temperature environment with a constant sliding distance of 1000 m while the applied force and sliding speed were varied. The load levels included 10, 20, and 30 N [41], while the sliding speeds ranged from 1 to 5 m/s. The pin and counter disc surfaces were prepared for each test by polishing them with silicon carbide abrasive grit papers with different grades (600, 1000, and 1500). Following this, the surfaces were cleaned with acetone to ensure their cleanliness. Before each wear test, the average roughness value of both the pin and counter disc surfaces was maintained at approximately 0.5 microns. The weight loss method, described in Equations (1)–(3) [42,43], was used to determine the volumetric wear rate, and the coefficient of friction was estimated by averaging the coefficient across a 1000 m sliding distance. Each wear test was repeated three times to minimise experimental errors, and the average values were reported. The worn pin surfaces were observed using SEM equipped with EDS.

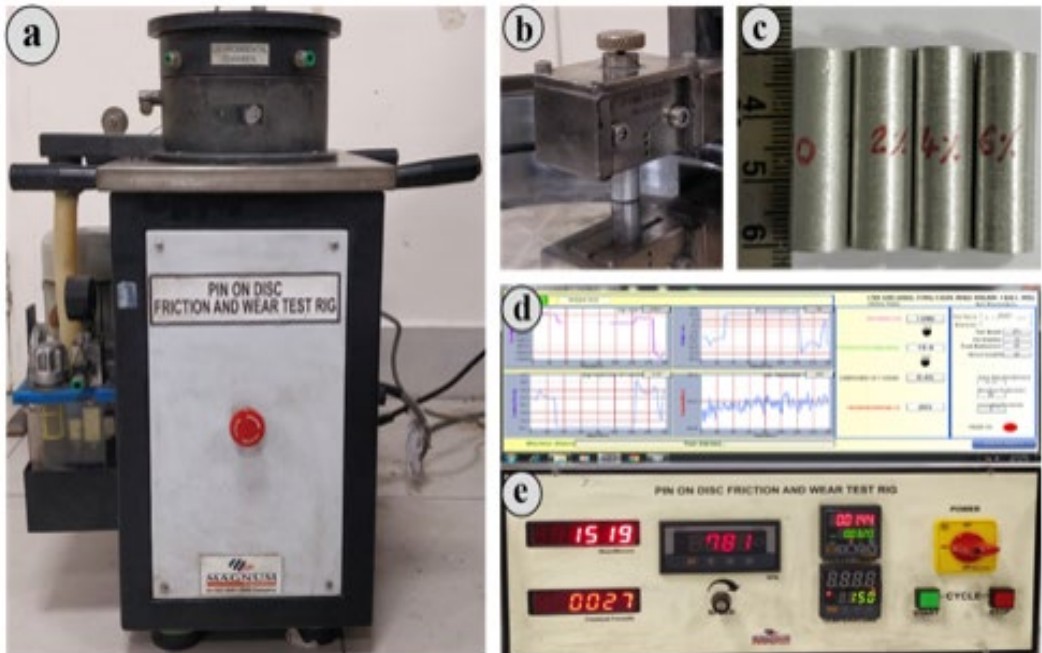

**Figure 3.** Dry sliding wear test: (**a**) Pin-on-disc setup, (**b**) a close-up of the pin holder, (**c**) pin samples, (**d**) real-time friction and wear graphs, and (**e**) machine controller.

$$\text{Mass loss of sample} = \frac{[\text{Initial Weight (g)} - \text{Final Weight (g)}]}{[\text{Density of the sample (g. mm}^{-3})]} \text{ mm}^3 \tag{1}$$

$$\text{Volumetric wear rate} = \frac{[\text{Mass loss of the sample}]}{[\text{Sliding distance (m)}]} \left(\frac{\text{mm}^3}{\text{m}}\right) \tag{2}$$

$$\text{Coefficient of Friction } (\mu) = \frac{[\text{Frictional force}]}{[\text{Applied load}]} \tag{3}$$

## 3. Results

### 3.1. Microstructure Analysis of AA8011-TiC Composite

The optical micrographs for the AA8011-TiC composites are shown in Figure 4a–d, illustrating different weight percentages of TiC. (Figure 4a–b) shows the microstructure of the as-cast AA8011-2 wt.% TiC, and the 4 wt.% TiC composite, revealing a relatively uniform dispersion of particulates. Although the presence of TiC particles is less prominent compared to the higher TiC content samples, the introduction of reinforcement contributes to the refinement of the microstructure. Notably, the size of the coarse grain in AA8011 is

reduced, indicating an improved microstructural refinement. Moving to (Figure 4c–d), the microphotographs present AA8011-TiC composites containing 6% and 8% TiC particles, respectively. These images demonstrate the consistent expansion of the ash-coloured region with increasing TiC concentration. No clustering or micro-pores were observed, indicating a homogeneous dispersion of TiC particles. Furthermore, increasing TiC content from 0% to 8% leads to a refinement of grain size from coarse to fine, consistent with previous studies [44]. The microstructure of the aluminium casting alloy with high titanium content is defined by the presence of dendrites during its solidification. In the spaces between these dendrites, one finds a composition of fine TiC particles and laths, originating from the eutectic mixture. As the molten mixture with high titanium content and TiC particles solidifies, the TiC particles are expelled towards the solid–liquid boundary and gather in the regions among the dendrites, forming the interdendritic areas. Consequently, this leads to a smaller grain size compared to castings without particles, and the TiC particles mix into the eutectic microstructure alongside Ti particles and laths [45].

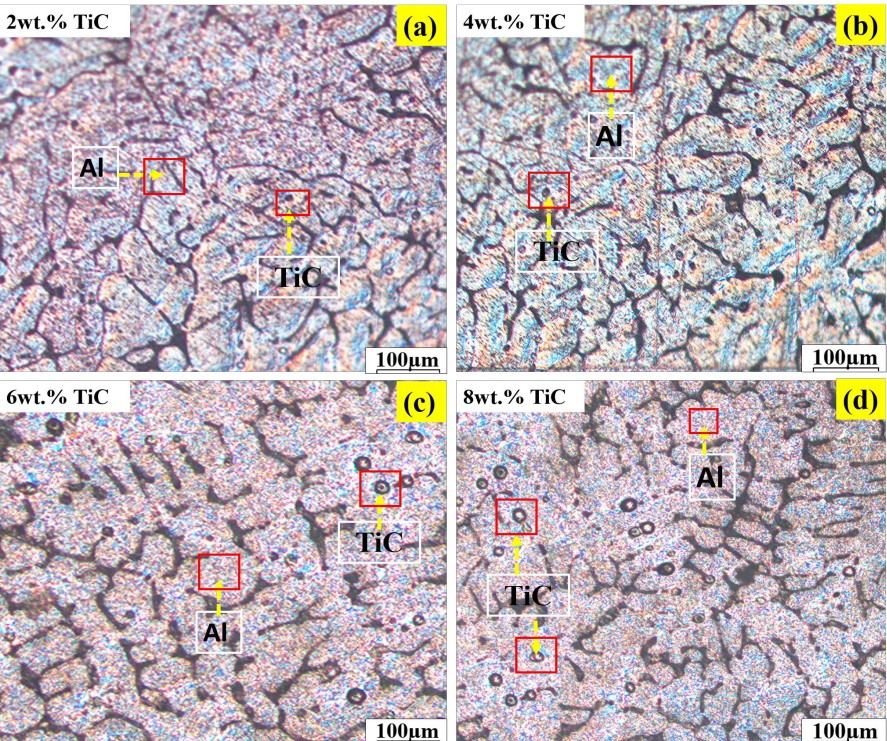

**Figure 4.** Optical microscope images of AA8011 alloy reinforced with varying amounts of TiC particles: (**a**) Al-2 wt.% TiC (**b**) Al-4 wt.% TiC, (**c**) Al-6 wt.% TiC, (**d**) Al-8 wt.% TiC.

Continuing the analysis to Figure 5, Figure 5a,b, presents SEM micrographs showing the as-cast AA8011 composites. The micrographs of the composites reveal a dendritic-type grain structure and the existence of TiC particles. The EDS spectrum of AA8011-2 wt.% TiC further confirms the existence of TiC particles within the composite material. Moving to Figure 5c,d, the microstructure of the TiC composite displays a dendritic morphology [46]. The inclusion of TiC particles contributes to the refinement of the grain structure, reducing the size of the large dendrites in the 4 wt.% TiC composite [47]. The EDS mapping of the matrix in Figure 5e,f, indicates the absence of TiC particles, whereas the 6 wt.% TiC composite samples reveal the presence of Ti and C elements. Furthermore, there are no signs of TiC particle aggregation in the composite samples, which can be attributed to the best selection of ultrasonic-assisted stir casting process parameters. Figure 5g,h, reveals the most significant improvement, where the dendritic microstructure is finely refined due to the higher number of individual TiC particles present in the 8 wt.% TiC sample. Figure 6a,b shows the spatial distribution and Elemental mapping of TiC in the 8 wt.% TiC composite

sample. The results of this study highlight the ability of the processing methodology used in this study to ensure a reasonably uniform dispersion of TiC particles within the matrix.

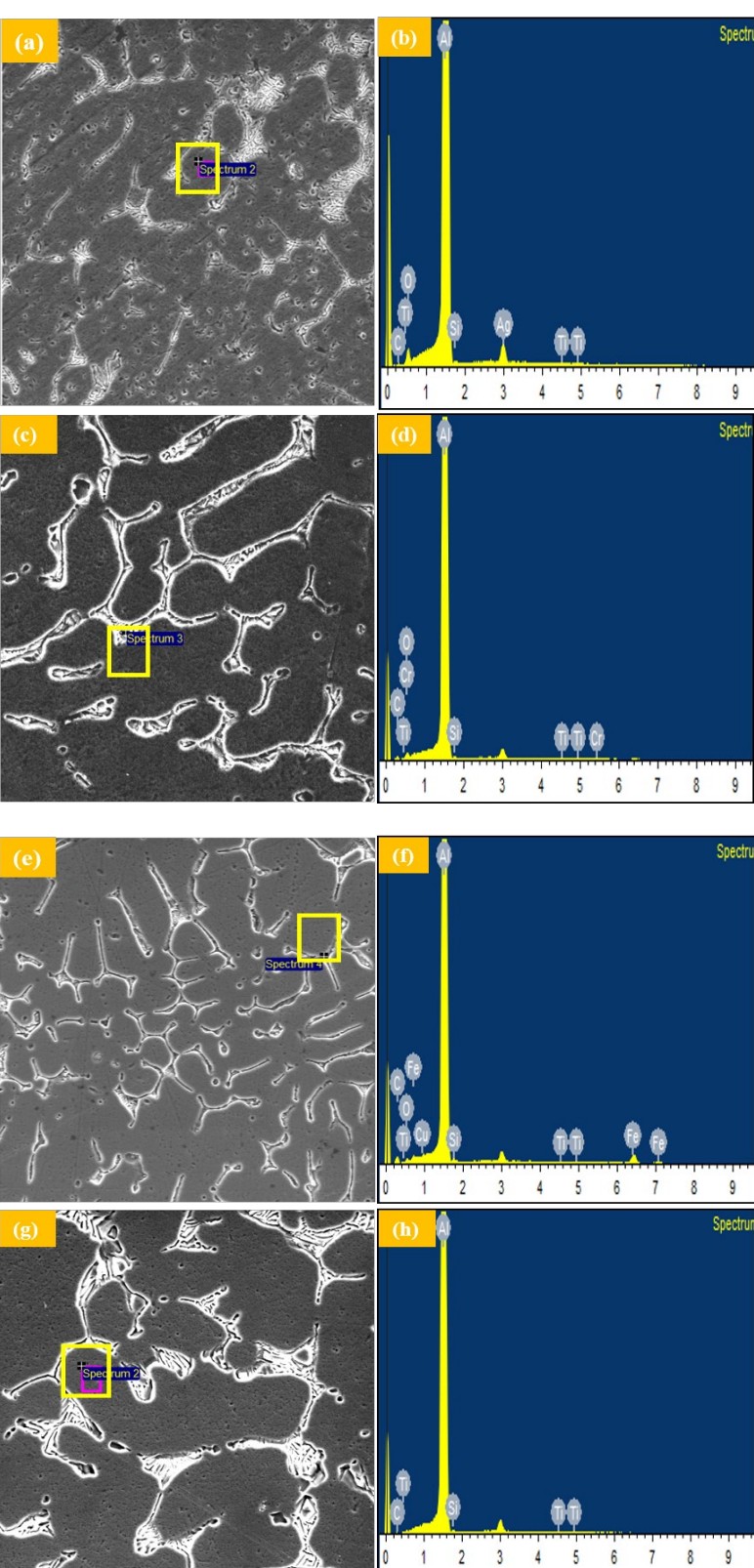

**Figure 5.** Scanning electron micrographs along with EDS mapping of the composites containing: (**a**,**b**) Al-2 wt.% TiC, (**c**,**d**) Al-4 wt.% TiC, (**e**,**f**) Al-6 wt.% TiC, (**g**,**h**) Al-8 wt.% TiC.

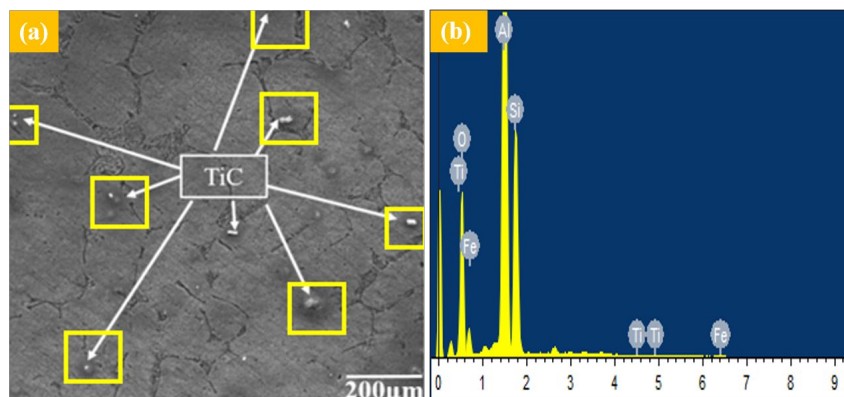

**Figure 6.** (**a**) SEM images depicting the spatial distribution of TiC particles in Al-8 wt.% TiC, (**b**) Elemental mapping showing the presence of major elements.

### 3.2. Analysis of AA8011-TiC Composite Materials Using X-ray Diffraction

The X-ray diffraction (XRD) pattern of the AA8011 aluminium alloy reinforced with TiC particulates is presented in Figure 7. The XRD pattern shows Al-TiC composites synthesised with different weight percentages (TiC 0–8 wt.%). The presence of aluminium is confirmed by the diffraction peaks observed at 38.55, 44.35, 65.30, and 78.35, corresponding to the Miller indices (111), (200), (220), and (311). The height of the diffraction peak in the XRD pattern for TiC is directly proportional to the content of ex situ-formed TiC particles in the composite material [48]. Therefore, increasing the number of TiC particles results in a corresponding increase in the peak height of the XRD pattern. In summary, in the XRD pattern of the Al-TiC composites the relationship between the peak height and the particulate content demonstrates the efficacy of the fabrication process in reinforcing the aluminium alloy with TiC particulates. Moreover, the absence of intermetallic compounds in the XRD pattern suggests that the resulting composite materials possess desirable properties suitable for various industrial applications [49].

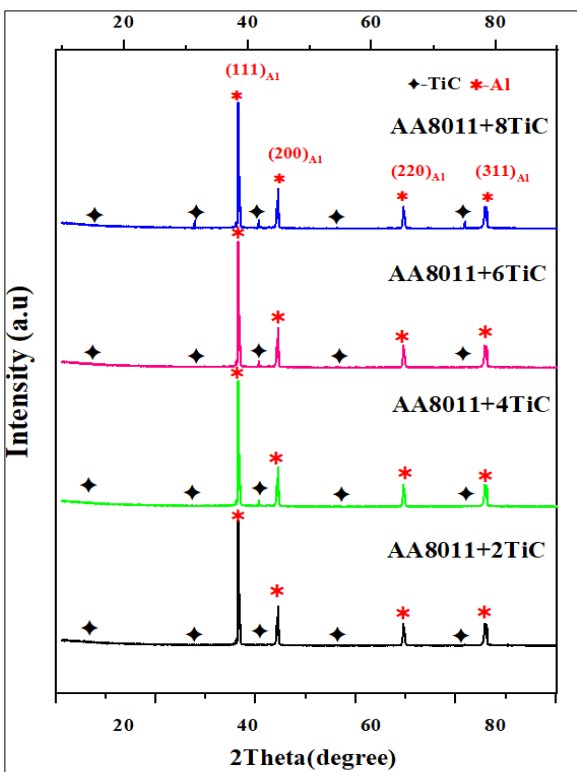

**Figure 7.** Results of X-ray diffraction on AA8011 as−cast and composites.

### 3.3. Micro−Hardness

Figure 8 shows the hardness measurement results. The base alloy AA8011 sample showed an average microhardness value of ~55.4 Vickers hardness (HV). With the incorporation of TiC particles, the hardness value was enhanced to 65.5 HV for the 2 wt.% TiC sample, ~72.46 HV for the 4 wt.% TiC sample, ~76.82 HV for 6 wt.% TiC sample, and a maximum of 80.2 HV for the 8 wt.% TiC specimen. The well-dispersed, harder TiC particles and their exceptional resistance to localised plastic deformation during indentation are the primary factors contributing to the significant improvement in microhardness. The microhardness value of the Al-TiC samples was much greater than the base alloy [50].

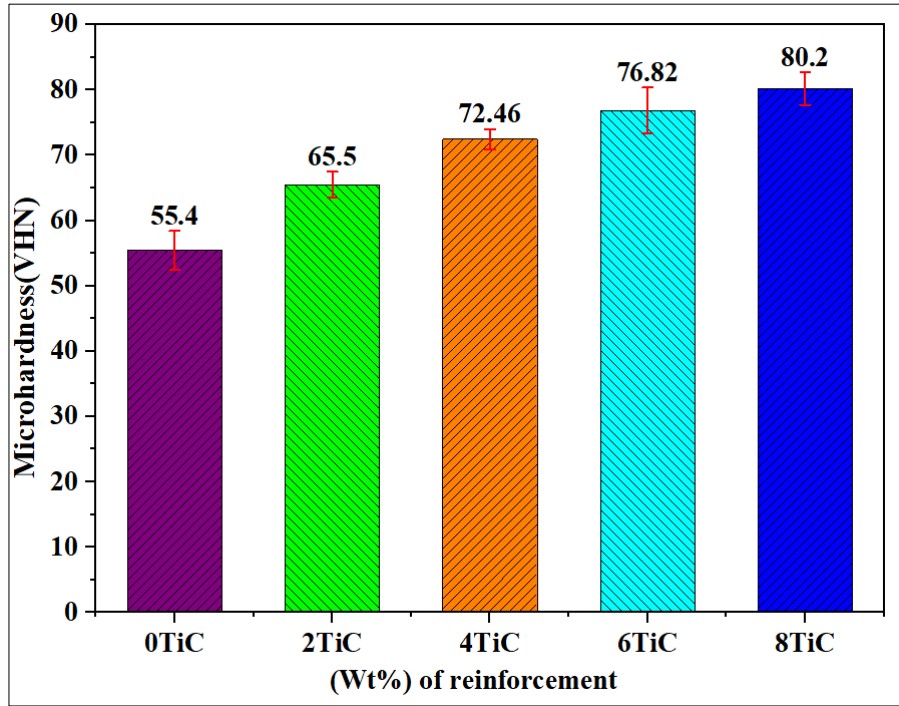

**Figure 8.** Micro hardness values of AA8011-TiC composites.

### 3.4. Density and Porosity

Figure 9 illustrates the density variations of AA8011-TiC MMCs with different TiC reinforcement percentages (0%, 2%, 4%, 6%, and 8%). Measured density, calculated density, and percentage of porosity were plotted against the weight percentage of particulates. It is demonstrates a positive correlation between TiC reinforcement and density values, indicating that an increase in TiC reinforcement leads to higher density values. It is important to note that the measured densities were slightly lower than the theoretical densities. The discrepancy may be attributed to the limited existence of casting defects in the MMCs, resulting in a small deviation from the expected densities. Nonetheless, the overall trend indicates that the incorporation of TiC reinforcement contributes to an enhancement in the density of the composites. The figure also demonstrates the influence of TiC particle composition on the percent porosity of the AA8011 matrix, with an increasing number of TiC particles leading to an increasing trend in % porosity. Equation (4) [51] was used to apply the Archimedes principle to determine the % porosity, with porosity variations attributed to several factors such as gas entrapment during mixing and shrinkage during solidification.

$$\text{Porosity}(\%) = [1 - \frac{(\text{Theoretical Density} - \text{Measured Density})}{(\text{Theoretical Density})}] \times 100\% \qquad (4)$$

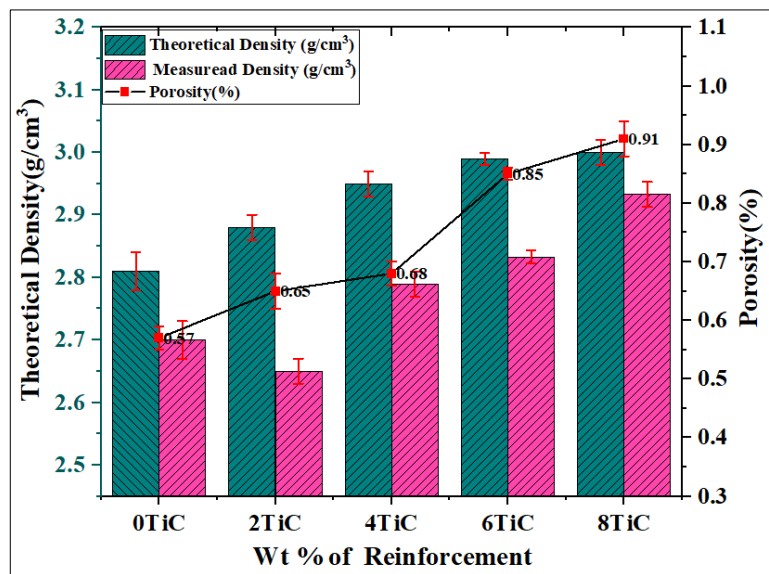

**Figure 9.** The correlation between experimental density, theoretical density (g/cm$^3$), and porosity (%) of Al−TiC composites.

### 3.5. Ultimate Tensile Strength and Elongation

Tensile test specimens were made from as-cast AMCs in accordance with the ASTM E08-8 standard. The stress–strain curves for the composites, depicting the variation observed with different weight fractions of reinforcement, are illustrated in Figure 10. The stress–strain curve analysis provides valuable information about the mechanical response and behaviour of the composites under tensile loading conditions. it is noticeable that all the TiC−reinforced composites have higher tensile strength than the AA8011 alloy [52]. The curves demonstrate the impact of TiC content on the mechanical behaviour of the composites. Furthermore, an increase in TiC content causes a shift of the stress–strain curves towards higher stress levels, indicating an improvement in strength due to the presence of the TiC reinforcement. However, the curves also reveal reduced deformation capacity as the TiC particles hinder the movement of dislocations and restrict plastic deformation.

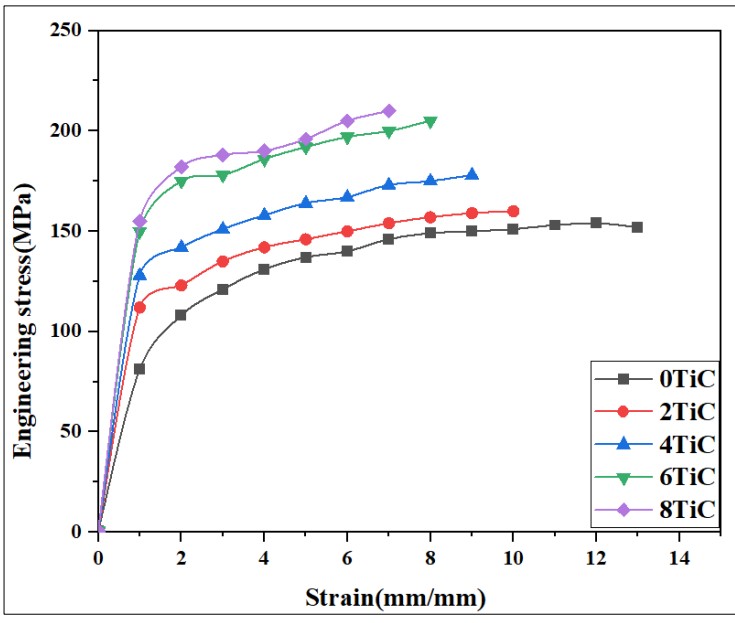

**Figure 10.** Analysis of stress–strain curves of Al-TiC composites with varying TiC contents.

The tensile strength values of Al-TiC composites are shown in Figure 11. It was noticed that increasing the number of TiC particles in the composite increased its tensile strength. This is owing to the excellent interfacial bonding between the matrix and the reinforcement, which distributes and transfers loads from the matrix to the reinforcement. Among the fabricated composites, it was found that the composite with 8% TiC particles displayed the highest strength. This suggests that incorporating TiC reinforcement improves the overall tensile strength of the composite material. The enhancement is due to the reinforcing effect of the hard TiC particles, which strengthen (Orowan strengthening plus load−bearing effects) the comparatively softer aluminium matrix and improve its tensile resistance. According to the Hall–Petch relationship, the increase in strength can also be attributed to microstructural refinement due to the incorporation of TiC reinforcement [53]. This suggests that as the TiC content increases, there is a compromise in ductility in the composite material [54,55]. The overall strength and ductility response of monolithic and composite samples is displayed in Figure 11.

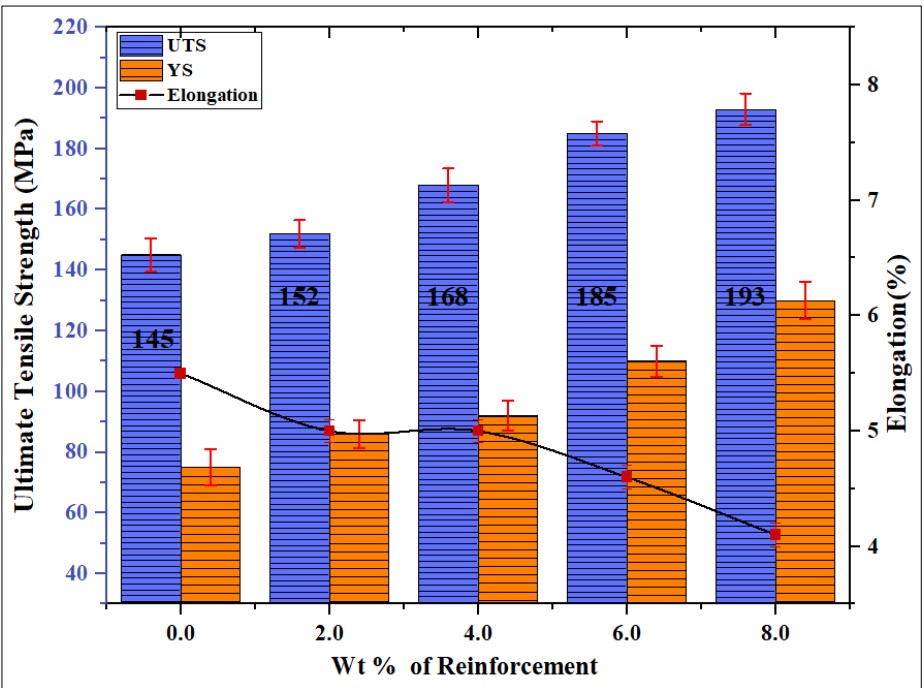

**Figure 11.** Effect of TiC addition on UTS, yield strength, and % elongation during tensile tests.

Figure 12 illustrates the results of SEM analysis conducted on the fractured surfaces of different specimens. In Figure 12a, the uniformly distributed large voids suggest that the fracture occurred in a ductile manner. This indicates significant plastic deformation and a ductile fracture of the unreinforced alloy. Moving to Figure 12b, the fractured surface of the Al-2 wt.% TiC composite exhibits medium and fine dimples, indicating reasonable plastic deformation. Figure 12c, displays the fractured surface of the Al-4 wt.% TiC composite, showing dimples of varying sizes, clusters of fine dimples, and reinforcement particles within the voids. The presence of deep voids with larger areas can be attributed to the detachment of reinforcement particles and the initiation and propagation of cracks caused by higher stress concentrations near the TiC particulates. Continuing to Figure 12d, the post-failure analysis of the fractured Al-6 wt.% TiC composite revealed a mixed ductile/brittle failure mode characterised by the presence of fine and large dimples, as well as elongated bands [56]. The development of voids within the metal matrix is a primary factor influencing the failure of MMCs. In Figure 12e, for the Al-8 wt.% TiC composites, a greater occurrence of cleavage and the presence of larger dimples indicated a brittle fracture failure.

Finally, Figure 12f, shows the presence of TiC reinforcement on the dendrites' surface, suggesting their presence in interdendritic regions.

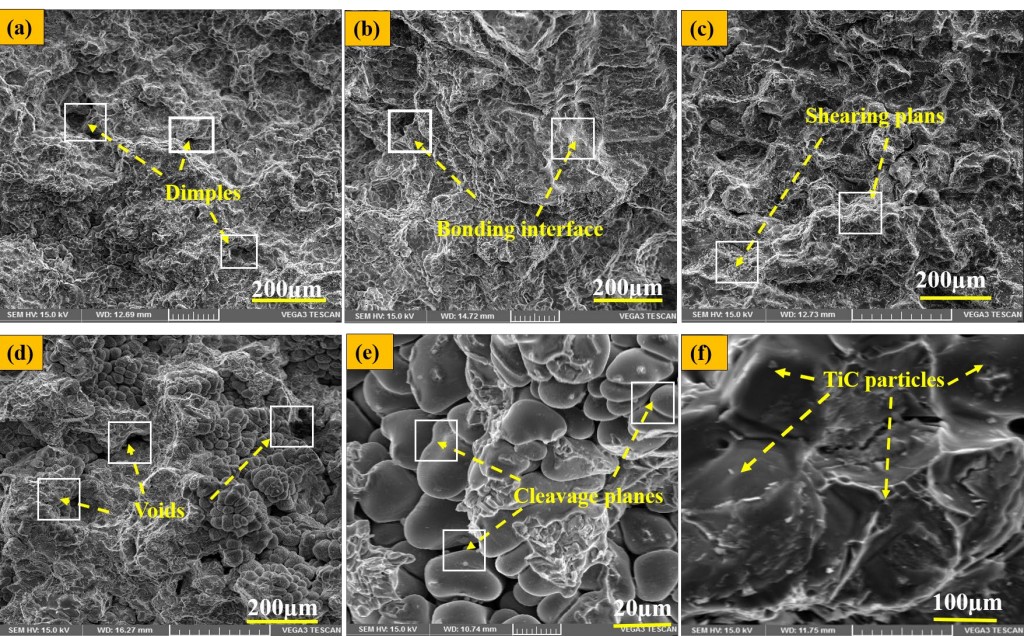

**Figure 12.** Scanning electron micrographs illustrating fracture morphologies of samples. (**a**) Al-0 wt.% TiC, (**b**) Al-2 wt.% TiC composite, (**c**) Al-4 wt.% TiC composite, (**d**) Al-6 wt.% TiC composite, (**e**) Al-8 wt.% TiC composite, and (**f**) enlarged view highlighting TiC particles covering the dendrites' surface (indicated by dotted yellow arrows).

## 4. Wear Rate

Figure 13 illustrates the rate of wear of the AA8011 alloy and Al-TiC composite under different loading conditions (10, 20, and 30 N) and different sliding velocities (1, 2, 3, 4, and 5 m/s) and at constant sliding distance 1000 m. It was found that the wear rate of the AA8011 alloy was higher than that of the synthesised Al-TiC composites. On the contrary, the wear rate of the composites displayed a contrasting trend. It demonstrated a linear increase up to a sliding velocity of 1–5 m/s, followed by a gradual decrease at high sliding velocities [22]. The wear rate at 1 m/s sliding velocity is displayed in Figure 13a, depicting the maximum wear rate observed in AA8011 and the minimum wear rate observed in the Al-6 wt.% TiC composite. As the sliding velocity increased, there was a notable rise in the amount of flash temperature generated between the pin surface and the counter disc. This led to the creation of a tribo layer that had an impact on the specific wear rate [57]. At lower sliding speeds (2–3 m/s) and various loads (10, 20, and 30 N), the wear rate of Al-TiC composites exhibited a linear increase, as shown in Figure 13b,c. This behaviour can be attributed to the longer duration of rubbing, leading to increased contact between the surfaces and higher material removal, resulting in elevated wear rates. Conversely, with higher sliding rates (4–5 m/s) and different loads (10, 20, and 30 N), a considerably increased wear rate was noticed, as depicted in Figure 13d,e. Several factors could contribute to this behaviour, including the formation of a protective oxide layer on the composite surface, changes in wear mechanisms, or the presence of surface debris that reduces direct metal-to-metal contact and lowers the wear rate [58]. Among the different volume percentages of TiC composites, the 6 wt.% TiC composite exhibited a relatively low wear rate. This is because the 8 wt.% TiC composites had a higher hardness than the other samples. The wear rate of the 6 wt.% TiC composite increased as the applied load increased while sliding at a speed of 2 m/s. This decline is mostly explained by the increased frictional heat produced at the pin's interaction with the counterface, which led

to the oxidation of the pin. As a consequence, there is minimal actual metal−to−metal contact between the pin and the counterface, leading to lower wear rates.

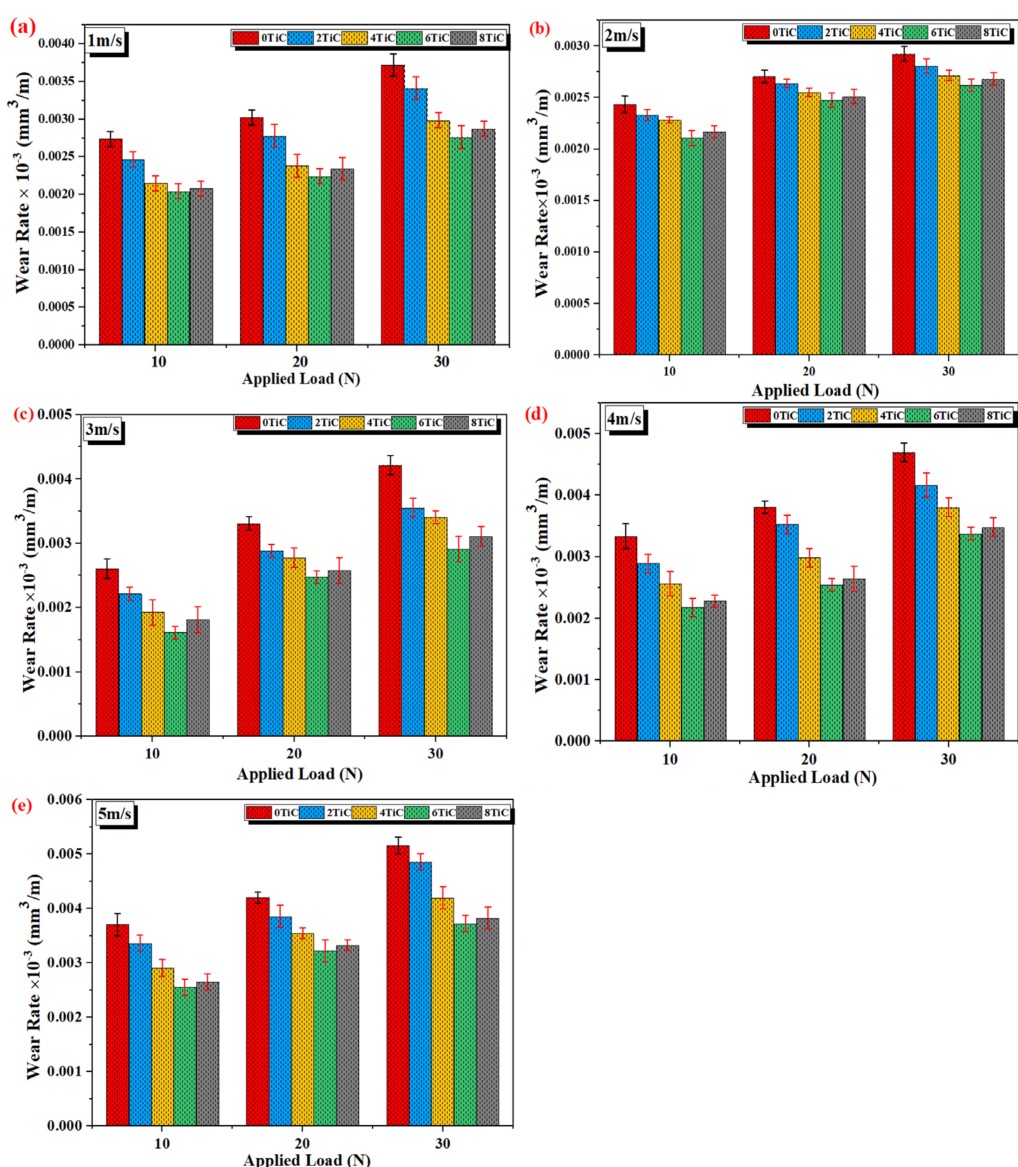

**Figure 13.** Wear rate as a function of applied load at different sliding velocities for alloy and composite: (**a**) 1 m/s, (**b**) 2 m/s, (**c**) 3 m/s, (**d**) 4 m/s, and (**e**) 5 m/s at a constant sliding distance of 1000 m.

## 4.1. Coefficient of Friction

Figure 14a–e, depicts the variations in the coefficient of friction (COF) of the AA8011 composite under varying weights of 10, 20, and 30 N at constant sliding distance 1000 m. As illustrated, the coefficient of friction falls as the applied load increases. According to the literature, this behaviour can be related to the production of an oxide layer at lower sliding speeds and a mechanically mixed layer at higher sliding rates of 5 m/s [59]. These layers effectively reduce the actual metallic contact and facilitate easy shearing of the contact interface. During the sliding motion of the pin and counterface, frictional heat is generated at their interface. This heat causes the formation of a mechanically mixed layer, which contains elements from both the pin and counterface materials. This layer is a result of the intense intermixing and deformation of the contacting surfaces. At lower sliding speeds, the formation of an oxide layer also occurs, further reducing the direct metallic contact between the surfaces. In the following sections, we will delve into a detailed discussion of

the formation and impact of both the oxide layer and the mechanically mixed layer on the frictional behaviour of the AA8011-TiC composite [60].

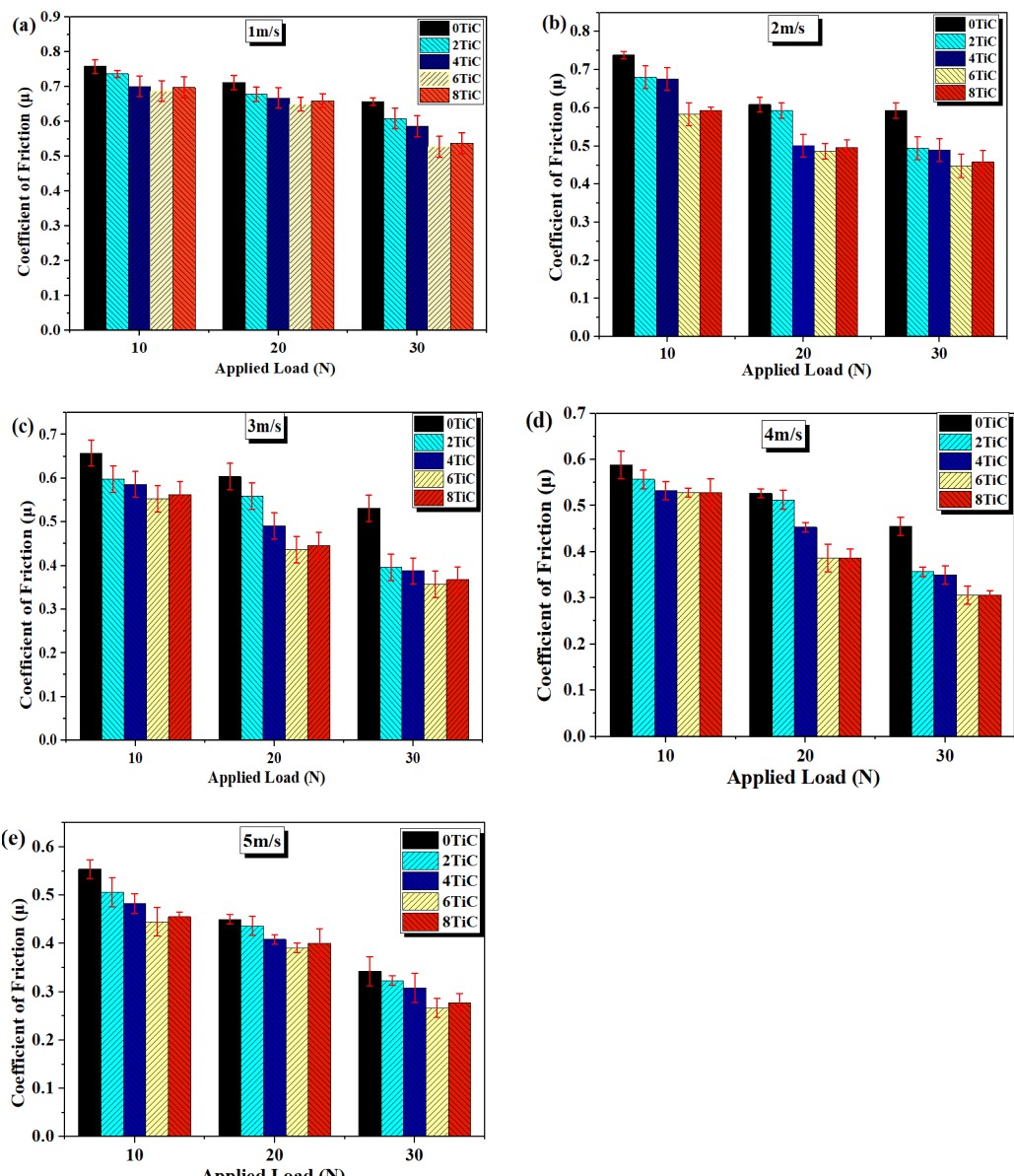

**Figure 14.** Plotting the coefficient of friction (COF) against the applied load at various sliding velocities for the base alloy and the composites. (**a**) 1 m/s, (**b**) 2 m/s, (**c**) 3 m/s, (**d**) 4 m/s, and (**e**) 5 m/s at a constant sliding distance of 1000 m.

### 4.2. Worn Pin Surface Analysis

Figure 15 depicts SEM images of worn pin samples investigated under low load conditions (10 N load and 1 m/s). Long grooves parallel to the sliding direction can be seen in all samples, indicating abrasive wear characteristics. However, there are notable distinctions between the base alloy and the TiC-reinforced composite samples. In the base alloy, deeper grooves accompanied by shallow craters are visible (Figure 15a–d) [61]. This effect is caused by the interaction of the soft pin surface and the hard counter surface, which results in low frictional heat and enhanced metallic contact. As a consequence, the hard asperities on the counter surface can penetrate deeper into the base alloy, leading to more pronounced surface damage. This behaviour aligns with the characteristics of delamination wear observed in the base alloy (Figure 15a,b). Delamination wear involves

the formation and propagation of subsurface cracks due to repeated sliding under applied loads. As these subsurface cracks propagate and merge, they eventually shear to the surface, resulting in material loss in the form of thin flakes and shallow craters or channels on the worn surface. Conversely, the presence of hard TiC particles within the aluminium alloy matrix adds to the deeper grooves observed in the SEM micrographs of TiC−reinforced composites. The TiC particles serve as abrasive agents, intensifying the abrasive wear mechanism. Additionally, the shallower craters observed in the base alloy samples indicate the occurrence of delamination wear. This wear mechanism involves the nucleation and propagation of subsurface cracks under repeated sliding, eventually leading to surface shearing and the formation of thin flakes and shallow craters or channels [62]. Thus, the wear surface analysis revealed distinct wear mechanisms between the base alloy and the TiC−reinforced composites, with delamination or fatigue wear being dominant in the base alloy and a combination of abrasion and oxidation prevailing in the TiC−reinforced composites [63].

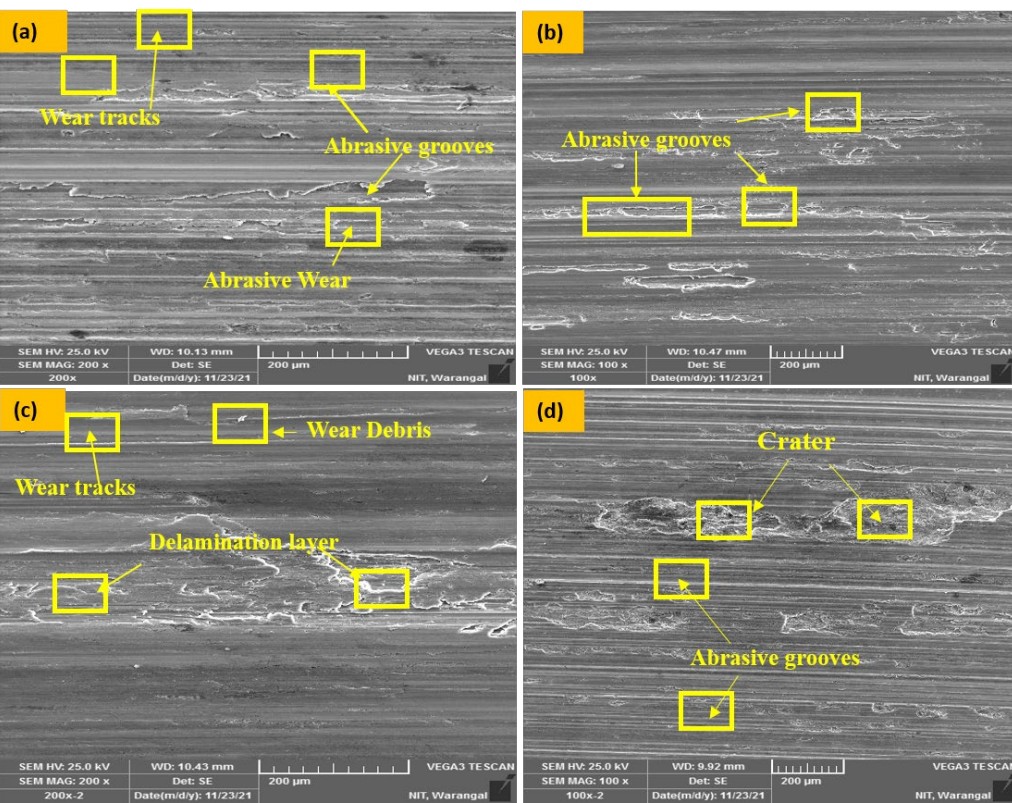

**Figure 15.** SEM micrographs show characteristics of abrasive wear in the sample examined at 10 N load and 1 m/s speed for (**a**) Al-2 wt.% TiC, (**b**) Al-4 wt.% TiC, (**c**) Al-6 wt.% TiC, and (**d**) Al-8 wt.% TiC.

Figure 16 shows SEM micrographs of the pin samples tested under a load of 20 N and a sliding speed of 3 m/s for all samples. It is noted that the worn pin surfaces of both the base alloys and composites exhibit thin white layers. Through EDS analysis, these spots/layers were identified as oxides, indicating the occurrence of oxidation wear (Figure 16a,b). The oxidation of the pin surface is a consequence of the heat generated during repeated sliding across the counter surface. As time progresses, the oxide debris accumulated in the valleys of the pin surface becomes compacted, leading to the formation of a protective surface oxide layer. The development of this oxide layer on the pin surface plays a crucial role in preventing direct metallic contact with the counter disc, consequently reducing the wear rate [41]. This is supported by the lower wear rates and coefficients of friction observed in the study. The existence of a mechanically mixed layer on the worn surface of the composite is also evident in (Figure 16c,d). This mechanically mixed layer

acts as a solid lubricant, promoting increased wear resistance and decreased friction. The presence of oxidation wear and the lubricating properties of the mechanically mixed layer are significant factors contributing to the enhanced wear resistance and reduced friction observed in the composites compared to the base alloys. These findings highlight the beneficial effects of the oxide layer and mechanically mixed layer on the wear response of the composite materials [64].

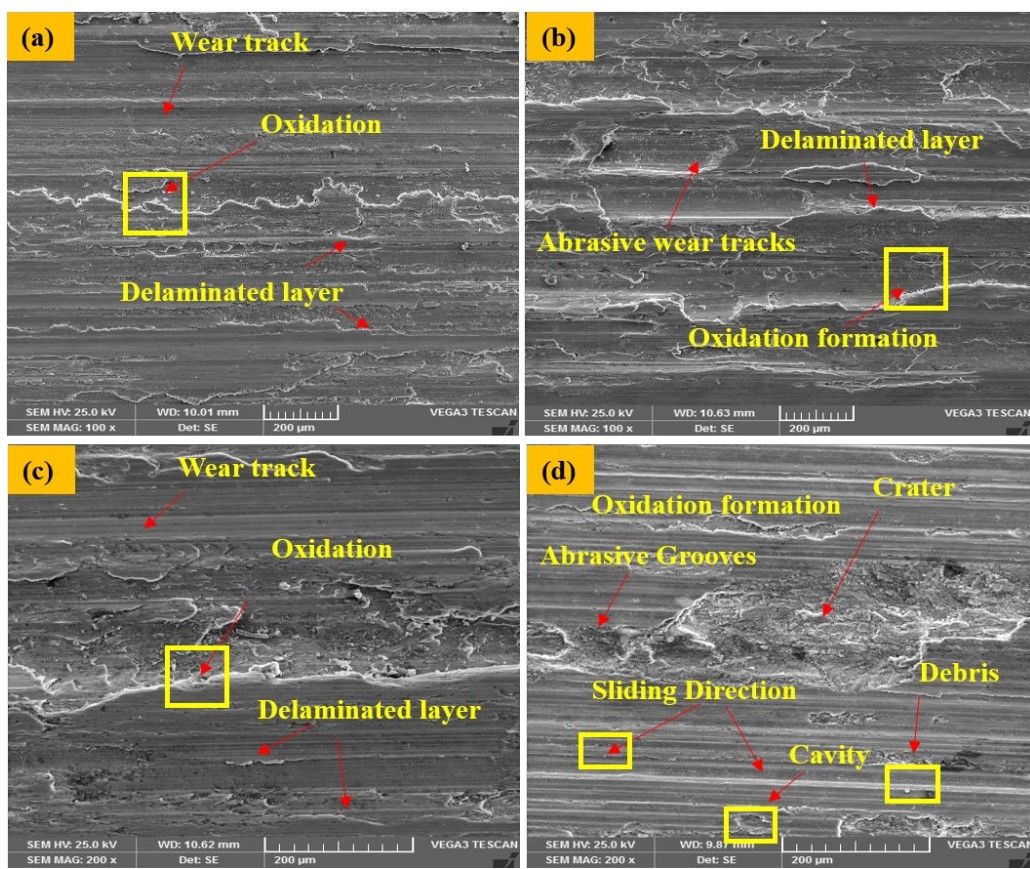

**Figure 16.** SEM/EDS micrographs show features of oxidation wear and corresponding elemental mapping in the sample examined at 20 N load and 3 m/s speed for (**a**) Al-2 wt.% TiC, (**b**) Al-4 wt.% TiC, (**c**) Al-6 wt.% TiC, and (**d**) Al-8 wt.% TiC.

Figure 17 shows EDS mapping analysis of the worn pin surface of the AA8011-6 wt.% TiC sample, which reveals the presence of iron (Fe) on the surface. This indicates that iron was detected and localised on the worn surface of the composite. The presence of iron may arise from various sources, such as wear debris, interactions between the pin and the counter surface, or iron−rich impurities present in the composite material. The EDS mapping results provide valuable information about the elemental distribution and composition of the worn surface, enhancing comprehension of the wear mechanisms and material responses exhibited by the Al-6 wt.% TiC composite.

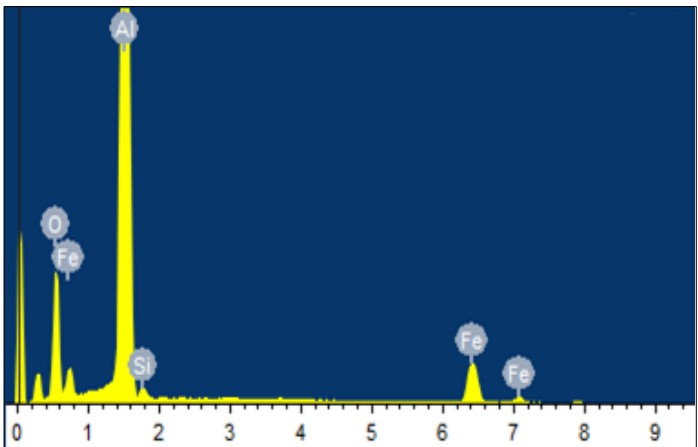

**Figure 17.** Elemental mapping showing the presence of Fe on the worn surface of the Al−6 wt.% TiC composite sample.

## 5. Conclusions

The influence of reinforcing TiC into aluminium alloy was studied through investigations, and the following conclusions were drawn.

- A stir casting technique with ultrasonic assistance was employed to fabricate composites with a uniform dispersion of TiC. The manufactured composite exhibits a higher density in comparison to the matrix, primarily due to the high density of TiC particles within its structure.
- TiC particle-reinforced composites were found to possess greater hardness compared to the unreinforced alloy. Among the composites reinforced with TiC particles, those containing 6% exhibited the highest hardness, measured at 61.5%.
- Incorporating TiC particles into the aluminium matrix improves the material's wear resistance. Under high load (30 N) and high-speed (5 m/s) conditions, composites containing 6% TiC particles achieved a maximum wear resistance of 32%.
- Incorporation of TiC particles into the aluminium matrix leads to a decrease in the friction coefficient. Notably, the inclusion of 6 wt.% TiC particles resulted in a significantly low coefficient of friction (0.266) under high load and speed conditions.
- Following a detailed examination of the worn surfaces, it was determined that the base alloy was worn predominantly due to abrasion and delamination. On the other hand, the AA8011 composite samples exhibited oxidation as the dominant wear mechanism.

**Author Contributions:** Conceptualization, C.B.G., M.B.P. and R.N.R.; methodology, C.B.G., M.B.P., S.I. and M.G.; validation, C.B.G., M.B.P. and R.N.R.; formal analysis, C.B.G., M.B.P. and S.I.; investigation, C.B.G. and M.B.P.; resources, R.N.R.; data curation, C.B.G. and M.B.P.; writing—original draft preparation, C.B.G. and M.B.P.; writing—review and editing, M.B.P., R.N.R., S.I. and M.G.; visualization, C.B.G., M.B.P. and S.I.; supervision, R.N.R. and S.I. All authors have read and agreed to the published version of the manuscript.

**Funding:** This research received no external funding.

**Data Availability Statement:** The raw/processed data required to reproduce these findings cannot be shared as the data also forms part of an ongoing study.

**Conflicts of Interest:** The authors declare no conflict of interest.

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
