# Peer review of "Influence of TiC Particles on Mechanical and Tribological Characteristics of Advanced Aluminium Matrix Composites Fabricated through Ultrasonic-Assisted Stir Casting"

_crystals, doi:10.3390/cryst13091360_

Round 1

Reviewer 1 Report

The paper can be accepted, but appropriate revisions are needed to enhance its overall quality.

·         Aluminum and aluminum alloys are finding increasing applications in various technical fields. They are employed in aviation, aerospace, military, automotive, and electronic industries. To improve the tribological and mechanical characteristics of aluminum alloys, different reinforcements are added, and corresponding composites are formed. Elaborate further on the application areas of aluminum and aluminum composites. Refer to papers: https://doi.org/10.1016/j.promfg.2016.12.045 , https://doi.org/10.1016/j.compositesb.2019.107329 , 10.1088/1742-6596/2212/1/012029 .

·         At the end of the introduction, specify the main contribution of the study and what sets it apart from similar papers s in the field. Why should this study be published?

·         Verify whether Table 1 provides information on the mass or volume content of elements.

·         As it deals with the mass content of reinforcements throughout the paper, unify the labeling of reinforcement content (wt.% or Wt % or Wt.%).

·         How many samples were used for testing the mechanical characteristics?

·         Considering its drawbacks compared to other casting methods, why was the stir casting process chosen?

·         Based on what criteria were appropriate parameters (load, sliding speed, sliding distance) chosen for forming the experimental plan? Which tribomechanical system do the working conditions correspond to?

·         How many different samples were used to test the tribological characteristics of the material? How many times was the experiment repeated?

·         To what extent do wear and friction coefficient results correlate with findings in the available literature? Provide explanations.

·         How much does the material's hardness change after testing? Was the hardness measured in the wear track?

·         Does the formation of MML layer occur?

·         What are the main wear mechanisms?

·         Based on the expanded analysis and discussion, expand the concluding remarks.

Reviewer 2 Report

Light microscopy is poor. The micrographs are out of focus. A lot of interdendritic porosity that will affect the mechanical properties. It is not described in the interdendritic spaces, which is eutéctico. The rest of the work is very relevant.

Reviewer 3 Report

The manuscript entitled “crystals-2599529” dealing with mechanical and tribological characteristics of advanced aluminium matrix has been reviewed. The paper has been nicely written but needs significant improvement. Please follow my comments.

1.     Figure 1 (B) is not redable. Please improve it.

2.     More discussion about this figure is required. Add more infor about SEM and setting.

3.     What is the main issue that will be solved by this investigation? Please clarify it in the text.

4.     Please add a brief statement on your methodology.

5.     Please proofread the paper.

6.     Aluminium matrix can be produced by different methods such as Additive Manufacturing. To highlight your work, add a short sentence about this note and refer to the following paper “Numerical and experimental investigations on manufacturability of Al–Si–10Mg thin wall structures made by LB-PBF”. “Microstructure and mechanical property considerations in additive manufacturing of aluminum alloys”.

Small work.
